# 5-Aryl-1-Arylideneamino-1*H*-Imidazole-2(3*H*)-Thiones: Synthesis and In Vitro Anticancer Evaluation

**DOI:** 10.3390/molecules26061706

**Published:** 2021-03-18

**Authors:** Ali H. Abu Almaaty, Eslam E. M. Toson, El-Sherbiny H. El-Sayed, Mohamed A. M. Tantawy, Eman Fayad, Ola A. Abu Ali, Islam Zaki

**Affiliations:** 1Zoology Department, Faculty of Science, Port Said University, Port Said 42526, Egypt; ali_zoology_2010@yahoo.com; 2Chemistry Department, Faculty of Science, Port Said University, Port Said 42526, Egypt; eslamatwa2010@gmail.com (E.E.M.T.); saeed201691@yahoo.com (E.-S.H.E.-S.); 3Medical Research Division, National Research Centre, Dokki, Giza 12511, Egypt; mohamed_tantawy@daad-alumni.de; 4Biotechnology Department, Faculty of Science, Taif University, P.O. Box 11099, Taif 21944, Saudi Arabia; e.esmail@tu.edu.sa; 5Chemistry Department, College of Science, Taif University, P.O. Box 11099, Taif 21944, Saudi Arabia; O.abuali@tu.edu.sa; 6Pharmaceutical Organic Chemistry Department, Faculty of Pharmacy, Port Said University, Port Said 42526, Egypt

**Keywords:** imidazole, synthesis, cytotoxicity, apoptosis, cell cycle analysis, Annexin V, VEGFR-2

## Abstract

A novel series of *N-1* arylidene amino imidazole-2-thiones were synthesized, identified using IR, ^1^H-NMR, and ^13^C-NMR spectral data. Cytotoxic effect of the prepared compounds was carried out utilizing three cancer cell lines; MCF-7 breast cancer, HepG2 liver cancer, and HCT-116 colon cancer cell lines. Imidazole derivative **5** was the most potent of all against three cell lines. DNA flow cytometric analysis showed that, imidazoles **4d** and **5** exhibit pre-G1 apoptosis and cell cycle arrest at G2/M phase. The results of the VEGFR-2 and B-Raf kinase inhibition assay revealed that compounds **4d** and **5** displayed good inhibitory activity compared with reference drug erlotinib.

## 1. Introduction

Cancer is the pathological uncontrolled proliferation of the abnormal cells that divide rapidly [1]. Moreover, cancer can involve any tissue of the body and have many different forms in each body area [2,3]. Therefore, there is a constant and growing interest to synthesize new biologically active molecules which might be useful in a cancer treatment.

Anticancer activity of the azole derivatives is most extensively studied and some of them are in clinical practice as antitumor active agents [4,5]. Imidazoles are an important class of heterocyclic molecules and imidazole derivatives are reported to have anticancer activity [6,7,8]. Misonidazole **I** displayed superior antiproliferative activity against a panel of cancer cell lines [9]. Tiazofurin **II** possesses good anticancer activity [10]. Lepidiline A and B are imidazole compounds which exhibit cytotoxicity against various types of human cancer cell lines at micromolar concentration [11]. Imidazole derivatives were found to exert their anticancer activity by acting mainly as antiangiogenic agents, inhibitors of B-Raf kinase, and as p38 MAP kinase inhibitors [12,13,14]. For example, compound **III** displayed potent antitumor activity against the MCF-7 cell line with an IC_50_ value of 3.26 μM, as compared with SOR (IC_50_ = 1.12 μM) [15]. Additionally, compound **IV** has been reported as a vascular endothelium growth factor (VEGFR-2) inhibitor and apoptotic inducer [15]. Moreover, it has been reported that imidazole molecule **V** inhibits the kinase activity of Raf in low nanomolecular concentrations [16]. Furthermore, compound **VI** was identified as a submicromolar inhibitor of B-Raf kinase [17] (Figure 1).

Based on the aforementioned aspects, the present study is concerned with the synthesis of novel imidazole molecules bearing arylidene amino substituents at the *N*-1 position. It is worthy to note that imidazole fragments are decorated at the C-5 position with phenyl groups carrying chloro or methoxy substituents to show the impact on anticancer activity. Finally, full details about the synthesis and evaluation of the antitumor activity in vitro are reported herein.

## 2. Result and Discussion

### 2.1. Chemistry

2-Arylhydrazinecarbothioamides **3a**–**e** were prepared in high yield according to the reported procedure [18,19] through condensation of aromatic aldehydes (namely, benzaldehyde, 2-hydroxybenzaldehyde, 5-bromo-2-hydroxybenzaldehyde, 4-methoxybenzaldehyde 4-hydroxy-3-methoxybenzaldehyde, and cinnamaldehyde) with thiosemicarbazide in refluxing ethanol containing catalytic amounts of acetic acid. The structure of **3a**–**e** was confirmed using different spectroscopic techniques. The ^1^H-NMR spectra revealed the presence of a signal for a azomethine group (CH=N) at δ 7.92–8.38 ppm. In addition to new signals attributed to NH and NH_2_ groups. Carbon signals observed in ^13^C-NMR spectra at δ 140.10–145.19 and 178.01–178.10 ppm assigned to carbons of azomethine (CH=N) and thiocarbonyl (C=S) groups, respectively, confirmed formation of 2-arylhydrazinecarbothioamides **3a**–**e**.

Furthermore, the novel imidazole derivatives **4a**–**e** and **5** were synthesized by cyclocondensation of 2-arylhydrazinecarbothioamides **3a**–**e** and substituted phenacylbromide in absolute ethanol in the presence of fused sodium acetate under reflux temperature (Scheme 1). The structure of novel imidazoles **4a**–**e** was characterized using different spectroscopic methods. The ^1^H-NMR spectra revealed the presence of additional aromatic signals attributed to phenyl ring B protons.

Compound **4b,** as a representative example, was established chemically via an acetylation reaction with acetic anhydride, which gave 2-((3-acetyl-5-(4-chlorophenyl)-2-thioxo-2,3-dihydro-1*H*-imidazol-1-ylimino)methyl)phenyl acetate (**6**), Scheme 1. The ^1^H-NMR spectrum of compound **6** showed the absence NH signals, and OH groups with the appearance of two new singlet signals at δ 2.08, 2.51, and 2.54 ppm due to the protons of two methyl function of acetyl groups. The ^13^C-NMR spectrum of compound **6** showed the presence of four new carbon signals at δ 20.02, 22.95, 169.50, and 172.13 related to two acetyl groups (CH_3_CO) which supported the formation of compound **6**.

### 2.2. In Vitro Antitumor Evaluation

#### 2.2.1. In Vitro Cytotoxic Activity against Three Cancer Cell Lines

The effect of imidazole molecules **4a**–**e** and **5** on the viability of three cancer cell lines were studied using the MTT assay. The cytotoxicity was assessed using Docetaxel (DOC) as a standard antitumor drug. The three cancer cell lines were MCF-7 (breast cancer cell line), colon carcinoma (HCT-116), and HepG2 (hepatocellular carcinoma cell line). Treatment of MCF-7, HCT-116, and HepG2 cell lines with different concentrations of imidazole derivatives revealed moderate to good antitumor activities as concluded from their IC_50_ values shown in Table 1. Compound 1-(2-hydroxybenzylideneamino)-5-(4-methoxyphenyl)-1*H*-imidazole-2(3*H*)-thione (**5**) was the most potent cytotoxic compound against all of the tested tumor cell lines with IC_50′_s < 5 μM. Moreover, the safety of compounds **4d** and **5** was assessed against a normal breast cell line (MCF-10A) using the MTT assay. The results revealed weak cytotoxic effect towards MCF-10A and compound **4d** showed less cytotoxic effect than standard drug DOC.

#### 2.2.2. Cell Cycle Analysis

The most active compounds, **4d** and **5** on MCF-7 were studied using DNA flow cytometric analysis. Treatment of MCF-7 cells with IC_50_ concentration dose value of compounds **4d** and **5** resulted in a significant alteration in cell cycle profile. There was a significant increase in the percentage of cell population at the G2/M phase from (8.79%) control to 35.55% and 25.63% upon treatment with compounds **4d** and **5**, respectively. Additionally, a significant increase in the percentage of cells at pre-G1 phase from 1.61% (control) to 22.36% and 22.81% upon treatment with compounds **4d** and **5**, respectively. Therefore, it can be concluded that compounds **4d** and **5** inhibit the cell proliferation of MCF-7 cells via cell cycle arrest at the G2/M phase and induction of apoptosis (Figure 2).

#### 2.2.3. Annexin V-FITC/PI Apoptosis Induction Analysis

Several imidazole molecules have been reported in the literature as apoptosis inducers in several tumor cell lines [20]. To evaluate the mode of cell death induced by compounds **4d** and **5**, Annexin V/FITC and propidium iodide (PI) assays were performed. MCF-7 cells were treated with compounds **4d** and **5** at their IC_50_ concentration dose value for 48 h. The results showed that the selected candidates induce apoptosis of MCF-7 cells at both early and later stages. As shown in Figure 3, the percent of early apoptosis was increased from 0.43% (control) to 6.12% and 2.66% for compounds **4d** and **5**, respectively. Additionally, the percent of late apoptosis was increased from 0.23% (control) to 10.23% and 13.64% upon exposure to compounds **4d** and **5**, respectively. The results proved that the antiproliferative activity of compounds **4d** and **5** is attributed to its apoptosis inducing activity in MCF-7 cells.

#### 2.2.4. VEGFR-2 Kinase Inhibitory Activity

To investigate the possible molecular mechanism underlying the potent anticancer activity of compounds **4d** and **5**. Their inhibitory activity against VEGFR-2 kinase was determined using an ELISA assay. The results in (Figure 4) showed that the selected imidazole molecules **4d** and **5** elicited potent activity against VEGFR-2 kinase with IC_50_ values 247.81 and 82.09 ng/mL, respectively, compared with erlotinib as the reference drug. According to these results, imidazole molecules **4d** and **5** possess good inhibition of VEGFR-2 kinase activity.

#### 2.2.5. B-Raf Ihibitory Activity

Raf is a protein kinase that initiates a cascade of other protein kinases by acting on the protein kinases MEK-1 and MEK-2 [21]. Therefore, constitutive activation of this signaling pathway is observed in variety of cancers. In the present assay, the activity potential of compounds **4d** and **5** was evaluated against B-Raf*^V600E^* using an ELISA assay. The results showed that compounds **4d** and **5** revealed good inhibitory activities against B-Raf*^V600E^* with IC_50_ values of 13.05 and 2.38 μg/mL, respectively, compared with the value of 0.98 μg/mL for erlotinib (Figure 5).

#### 2.2.6. Molecular Docking Study

Compounds **4d** and **5** were docked into the VEGFR-2 crystal structure (PDB code: 3U6J) [22]. Compound **4d** interacted with the active site of 3U6J by hydrogen bonding with the amino acid Lys 868 by its methoxy group. In addition, the hydrophobicity induced by compound **4d** resulted in a docking score of −20.69 kcal/mol. On the other hand, compound **5** interacted through hydrogen bonding with the amino acids Lys 868 and Ile 1025 by its methoxy and hydroxy groups, respectively, and its hydrophobicity resulted in docking score of −21.58 kcal/mol (Figure 6). The interaction result is consistent with the obtained IC_50_ values against the VEGFR-2 enzyme. Accordingly, from the obtained results, compound **5** was more potent than compound **4d** and with higher VEGFR-2 inhibitory activity.

## 3. Material and Methods

### 3.1. General

^1^H and ^13^C-NMR spectra were measured with a Bruker 400 DRX, Avance NMR spectrometer (Chichago, Elk Grove Village, USA) using the DMSO-*d_6_* solvent. The IR data were obtained with a shimadzu 470 spectrometer (Kyoto, Kyoto, Japan). Melting points of the prepared derivatives were measured with an electro thermal melting point apparatus and were not corrected. The microanalyses were implemented on a Perkin Elmer CHN elemental analyzer (Haan, Germany). All chemicals were purchased from Aldrich chemical company and all reagents were of analytical grade. The prepared imidazole molecules were soluble in DMSO and not soluble in water at any concentration.

### 3.2. General Procedure for the Synthesis of 2-Arylhydrazinecarbothioamide (**3a–d**)

To a mixture of aromatic aldehyde (0.01 mol) and thiosemicarbazide **2** in absolute ethanol (50 mL) and a few drops of glacial acetic acid were added. The reaction mixture was heated under reflux for 4 h. After cooling, the separated solid was collected by filtration, dried, and crystallized from ethanol to give **3a**–**d**.

#### 3.2.1. (E)-2-Benzylidenehydrazinecarbothioamide (**3a**)

Pale yellow crystals, yield 83%. m.p. 188–190 °C. IR (KBr) ν_max_: 3418, 3262, 3154 (NH), 3072 (arom.CH), 1591 (C=N), 1536, 1451 (C=C), 1292 (C=S) cm^–1^. ^1^H-NMR (DMSO-*d_6_*) δ: 7.51–7.35 (m, 3H, arom. CH), 7.81 (dd, *J* = 6.4, 3.0 Hz, 2H, arom. CH), 8.03 (s, 1H, NH), 8.06 (s, 1H, CH=N), 8.24 (s, 1H, NH), 11.47 (s, 1H, NH) ppm. ^13^C-NMR (DMSO-*d_6_*) δ: 125.52, 127.40, 129.32, 139.34 (C- aromatic), 145.19 (CH=N), 178.10 (C=S) ppm.

#### 3.2.2. (E)-2-(2-Hydroxybenzylidene)hydrazinecarbothioamide (**3b**)

Yellow crystals, yield 81%, m.p. 210–212 °C. IR (KBr) ν_max_: 3444 (OH), 3319, 3140 (NH), 3032 (arom.CH), 1613 (C=N), 1539, 1491 (C=C), 1368 (C=S), 1113 (C–O) cm^−1^. ^1^H-NMR (DMSO-*d_6_*) δ: ^1^H-NMR (DMSO-*d_6_*) δ: 6.80 (t, J = 8.4 Hz, 1H, arom.CH), 6.86 (d, J = 8.4 Hz, 1H, arom. CH), 7.20 (t, J = 8.5 Hz, 1H, arom.CH), 7.91 (s, 2H, arom.CH and NH), 8.15 (s, 1H, CH=N), 8.38 (s, 1H, NH), 9.90 (br. s, 1H, OH), 11.38 (s, 1H, NH) ppm. ^13^C-NMR (DMSO-*d_6_*) δ: 116.50, 119.75, 120.80, 127.21, 131.57 (C-aromatic), 140.10 (CH=N), 156.87 (C–O), 178.10 (C=S) ppm.

#### 3.2.3. (E)-2-(4-Methoxybenzylidene)hydrazinecarbothioamide (**3c**)

Yellow crystals, yield 81%, m.p. 192–194 °C. IR (KBr) ν_max_: 3404, 3290, 3154 (NH), 1604 (C=N), 1536, 1511 (C=C), 1361 (C=S) cm^−1^. ^1^H-NMR (DMSO-*d_6_*) δ: 3.78 (s, 3H, OCH_3_), 6.96 (d, *J* = 8.6 Hz, 2H, arom.CH), 7.74 (d, *J* = 8.6 Hz, 2H, arom.CH), 7.95 (s, 1H, NH), 8.02 (s, 1H, CH=N), 8.16, 11.37 (s, 2H, NH_2_) ppm. ^13^C-NMR (DMSO-*d_6_*) δ: 55.73 (OCH_3_), 114.60, 127.17, 129.38 (C-aromatic), 142.72 (CH=N), 161.12 (C–O), 178.01 (C=S) ppm.

#### 3.2.4. (E)-2-((E)-3-Phenylallylidene)hydrazinecarbothioamide (**3d**)

Pale yellow crystals, yield 79 %, m.p. 134–136 °C. IR ν_max_: 3418, 3262, 3154 (NH), 3027 (arom.CH), 1591 (C=N), 1536, 1451 (C=C), 1371 (C=S) cm^–1^. ^1^H-NMR (DMSO-*d_6_*) δ: 6.88 (dd, *J* = 16.1, 9.2 Hz, 1H olefinic CH), 7.03 (d, *J* = 16.1 Hz, 1H, olefinic CH), 7.45–7.34 (m, 3H, arom.CH), 7.57 (d, *J* = 7.2 Hz, 2H, arom.CH), 7.68 (s, 1H, NH), 7.92 (d, *J* = 9.2 Hz, 1H, CH=N), 8.22, 11.44 (s, 2H, NH_2_) ppm. ^13^C-NMR (DMSO-*d_6_*) δ: 119.12, 125.52, 127.40, 129.32, 136.32, 139.34 (C-aromatic and olefinic), 145.19 (CH=N), 178.10 (C=S) ppm.

### 3.3. General Procedure for the Synthesis of Imidazole Derivatives **4a–e** and **5**

A suspension of thiosemicarbazone derivative **3a**–**e** (0.01 mol) and freshly prepared substituted phenacyl bromide (0.01 mol) in 30 mL of absolute ethanol and freshly fused sodium acetate (0.03 mol) was refluxed for 6 h. After concentration and cooling, the solid that formed was collected and crystallized from proper solvent to give the target imidazole compound.

#### 3.3.1. (E)-1-(Benzylideneamino)-5-(4-chlorophenyl)-1H-imidazole-2(3H)-thione (**4a**)

Pale yellow crystals, yield 78%, m.p. 121–123 °C crystallized from dioxane; IR (KBr) ν_max_: 3403 (NH), 3109 (arom.CH), 1604 (C=N), 1566, 1512 (C=C), 1255 (C=S) cm^−1^. ^1^H-NMR (DMSO-*d_6_*) δ: 7.32 (s, 1H, H-4 of imidazole ring), 7.48–7.38 (m, 5H, arom.CH), 7.64 (d, *J* = 7.1 Hz, 2H, arom.CH), 7.85 (d, *J* = 8.5 Hz, 2H, arom.CH), 8.04 (s, 1H, CH=N), 12.12 (s, 1H, NH) ppm. ^13^C-NMR (DMSO-*d_6_*) δ: 104.93 (C4 imidazole), 126.71 (C2,6 chlorophenyl), 127.67 (C3,5 chlorophenyl), 129.10 (C3,5 phenyl), 129.33 (C2,6 phenyl), 129.87 (C4 imidazole), 132.45 (C4 phenyl), 133.84 (C1 chlorophenyl), 134.68 (C4 chlorophenyl), 142.02 (C4 phenyl), 149.75 (C=N), 178.27 (C=S) ppm. Anal. Calcd for C_16_H_12_ClN_3_S (313.80). Calculated: C, 61.24; H, 3.85; N, 13.39. Found: C, 61.12; H, 3.61; N, 13.48.

#### 3.3.2. (E)-5-(4-Chlorophenyl)-1-(2-hydroxybenzylideneamino)-1H-imidazole-2(3H)-thione (**4b**)

Yellow crystals, yield 79 %, m.p. 128–130 °C; crystallized from ethanol/ H_2_O (3:1); IR (KBr) ν_max_: 3442 (OH), 3315 (NH), 3039 (arom.CH), 1614 (C=N), 1578. 1540 (C=C), 1267 (C=S), 1092 (C–O) cm^–1^. ^1^H-NMR (DMSO-d_6_) δ: 6.95–6.88 (m, 2H, arom.CH), 7.29–7.18 (m, 2H, arom.CH), 7.40 (s, 1H, H-4 of imidazole ring), 7.48 (d, J = 8.5 Hz, 2H, arom.CH), 7.88 (d, J = 8.5 Hz, 2H, arom.CH), 8.35 (s, 1H, CH= N), 10.14 (s, 1H, OH), 12.18 (s, 1H, NH) ppm. ^13^C-NMR (DMSO-d_6_) δ: 104.64 (C4 imidazole), 116.61 (C3 hydroxyphenyl), 119.08 (C1 hydroxyphenyl), 120.83 (C5 hydroxyphenyl), 127.71 (C2,6 chlorophenyl), 129.12 (C3,5 chlorophenyl), 131.06 (C6 hydroxyphenyl), 131.55 (C5 imidazole), 132.43 (C1 chlorophenyl), 133.93 (C4 hydroxyphenyl), 139.98 (C4 chlorophenyl), 149.76 (C=N), 156.41 (C2 hydroxyphenyl), 178.08 (C=S) ppm. Anal. Calcd for C_16_H_12_ClN_3_OS (329.80). Calculated: C, 58.27; H, 3.67; N, 12.74. Found: C, 58.41; H, 3.89; N, 12.49.

#### 3.3.3. (E)-1-(5-Bromo-2-hydroxybenzylideneamino)-5-(4-chlorophenyl)-1H-imidazole-2(3H)-thione (**4c**)

Yellow crystals, yield 78%, m.p. 191–193 °C; crystallized from ethanol/ H_2_O (3:1); IR (KBr) ν_max_: 3454 (OH), 3252 (NH), 3059 (arom.CH), 1592 (C=N), 1548, 1477 (C=C), 1263 (C=S), 1086 (C–O) cm^−1^. ^1^H-NMR (DMSO-*d_6_*) δ: 6.89 (d, *J* = 8.7 Hz, 1H, arom.CH), 7.35 (d, *J* = 8.7 Hz, 1H, arom.CH), 7.41 (s, 1H, H-4 of imidazole ring), 7.47 (d, *J* = 8.5 Hz, 2H, arom.CH), 7.77 (s, 1H, arom.CH), 7.88 (d, *J* = 8.5 Hz, 2H, arom.CH), 8.27 (s, 1H, CH=N), 10.41 (s, 1H, OH), 12.29 (s, 1H, NH) ppm. ^13^C-NMR (DMSO-*d_6_*) δ: 104.93 (C4 imidazole), 111.29 (C5 bromohydroxyphenyl), 118.87 (C3 bromohydroxyphenyl), 123.19 (C1 bromohydroxyphenyl), 127.70 (C2,6 chlorophenyl), 129.11 (C3,5 chlorophenyl), 132.45 (C5 imidazole and C6 bromohydroxyphenyl), 133.15 (C1,4 chlorophenyl), 137.50 (C4 bromohydroxyphenyl), 149.85 (C=N), 155.47 (C2 bromohydroxyphenyl), 178.28 (C=S) ppm. Anal. Calcd for C_16_H_11_BrClN_3_OS (408.70). Calculated: C, 47.02; H, 2.71; N, 10.28. Found: C, 46.89; H, 2.57; N, 10.43.

#### 3.3.4. (E)-5-(4-Chlorophenyl)-1-(4-methoxybenzylideneamino)-1H-imidazole-2(3H)-thione (**4d**)

Yellow crystals, yield 81%, m.p. 151–153 °C; crystallized from ethanol/H_2_O (3:1); IR (KBr) ν_max_: 3422 (NH), 3033 (arom.CH), 1575 (C=N), 1477, 1426 (C=C), 1352 (C=S), 1092 (C–O) cm^–1^. ^1^H-NMR (DMSO-*d_6_*) δ: 3.78 (s, 3H, OCH_3_), 6.98 (d, *J* = 8.8 Hz, 2H, arom.CH), 7.34 (s, 1H, H-4 of imidazole ring), 7.44 (d, *J* = 8.6 Hz, 2H, arom.CH), 7.59 (d, *J* = 8.8 Hz, 2H, arom.CH), 7.86 (d, *J* = 8.6 Hz, 2H, arom.CH), 7.99 (s, 1H, C=N), 12.04 (s, 1H, NH) ppm. ^13^C-NMR (DMSO-*d_6_*) δ: 55.71 (OCH_3_), 104.621 (C4 imidazole), 114.80 (C3,5 methoxyphenyl), 127.43 (C1 methoxyphenyl), 127.66 (C2,6 chlorophenyl), 128.29 (C3,5 chlorophenyl), 129.07 (C2,6 methoxyphenyl), 132.34 (C5 imidazole), 134.05 (C1 chlorophenyl), 141.88 (C4 chlorophenyl), 142.68 (CH=N), 160.72 (C4 methoxyphenyl), 178.25 (C=S) ppm. Anal. Calcd for C_17_H_14_ClN_3_OS (343.83). Calculated: C, 59.38; H, 4.10; N, 12.22. Found: C, 59.29; H, 3.89; N, 12.06.

#### 3.3.5. 5-(4-Chlorophenyl)-1-((E)-((E)-3-phenylallylidene)amino)-1H-imidazole-2(3H)-thione (**4e**)

Pale yellow crystals, yield 78%, m.p. 139–141 °C crystallized from dioxane; IR (KBr) ν_max_: 3499 (NH), 3059 (arom.CH), 1683 (C=N), 1565, 1480 (C=C), 1259 (C=S) cm^−1^. ^1^H-NMR (DMSO-*d_6_*) δ: 6.94 (s, 1H, H-4 of imidazole ring), 7.02–7.04 (m, 1H, olefinic CH), 7.30 (d, *J* = 7.3 Hz, 1H, olefinic CH), 7.38–7.32 (m, 3H, arom.CH), 7.46 (d, *J* = 8.6 Hz, 2H, arom.CH), 7.58 (d, 2H, arom.CH), 7.88 (d, *J* = 10.1, 3.3 Hz, 2H, arom.CH), 7.93 (d, *J* = 4.1 Hz, 1H, CH=N), 12.16 (s, 1H, NH) ppm. ^13^C-NMR (DMSO-*d_6_*) δ: 104.81 (C4 imidazole), 125.63 (C2 allylidene), 127.36 (C2,6 chlorophenyl), 127.68 (C2,6 phenyl), 129.08 (C3,5 phenyl), 129.23 (C3,5 chlorophenyl), 130.49 (C4 phenyl), 132.45 (C5 imidazole), 133.98 (C1 chlorophenyl), 136.60 (C3 allylidene), 137.22 (C4 chlorophenyl), 144.59 (C1 phenyl), 149.80 (C=N), 178.16 (C=S) ppm. Anal. Calcd for C_18_H_14_ClN_3_S (339.84). Calculated: C, 63.62; H, 4.15; N, 12.36. Found: C, 63.53; H, 4.01; N, 12.12.

#### 3.3.6. (E)-1-(2-Hydroxybenzylideneamino)-5-(4-methoxyphenyl)-1H-imidazole-2(3H)-thione (**5**)

Yellow crystals, yield 76.40%, m.p. IR (KBr) ν_max_: 3442 (OH), 3319 (NH), 3032 (arom.CH), 2986 (aliph.CH), 1613 (C=N), 1540, 1489 (C=C), 1271 (C=S), 1203, 1056 (C–O) cm^−1^. ^1^H-NMR (DMSO-*d_6_*) δ: 3.89 (s, 3H, OCH_3_), 6.92–6.80 (m, 3H, arom.CH), 7.17 (d, *J* = 8.8 Hz, 1H, arom.CH), 7.22 (d, *J* = 8.8 Hz, 1H, arom.CH), 7.31 (s, 1H, H-4 of imidazole ring), 7.64 (d, *J* = 6.7 Hz, 1H, arom.CH), 7.94 (d, *J* = 6.3 Hz, 2H, arom.CH), 8.39 (s, 1H, CH=N), 9.91 (s, 1H, OH), 11.41 (s, 1H, NH) ppm. ^13^C-NMR (DMSO-*d_6_*) δ: 56.76 (OCH_3_), 102.99 (C4 imidazole), 111.25 (C3 hydroxyphenyl), 113.20 (C1 hydroxyphenyl), 116.47 (C5 hydroxyphenyl), 119.72 (C3,5 methoxyphenyl), 127.17 (C1 methoxyphenyl), 130.41 (C6 hydroxyphenyl), 131.05 (C5 imidazole), 131.55 (C2,6 methoxyphenyl), 139.97 (C4 hydroxyphenyl), 155.26 (C=N), 156.41 (C2 hydroxyphenyl), 156.86 (C4 methoxyphenyl), 178.06 (C=S) ppm. Anal. Calcd for C_17_H_15_N_3_O_2_S (325.38). Calculated: C, 62.75; H, 4.65; N, 12.91. Found: C, 62.51; H, 4.78; N, 13.08.

### 3.4. General Procedure for the Synthesis (E)-2-((3-Acetyl-5-(4-chlorophenyl)-2-thioxo-2,3-dihydro-1H-imidazol-1-ylimino)methyl)phenyl Acetate (**6**)

A solution of compound **4b** (0.01 mol) in acetic anhydride (20 mL) was heated under reflux for 2 h, then cooled down and decanted into water. The reaction mixture was left for 24 h, and the precipitate formed was filtered off, washed with water and dried. Finally, the product was crystallized from benzene to give **6**.

Colorless crystals, yield 63%, m.p. 115–117 °C. IR (KBr) ν_max_: 3090 (arom.CH), 2934 (aliph.CH), 1766, 1696 (C=O), 1606 (C=N), 1521, 1484 (C=C), 1274 (C=S) cm^−1^. ^1^H-NMR (DMSO-d_6_) δ: 2.08 (s, 3H, COCH_3_), 2.54 (s, 3H, COCH_3_), 7.29 (d, J = 8.1 Hz, 1H, arom.CH), 7.44 (t, J = 7.5 Hz, 1H, arom.CH), 7.52 (d, J = 8.5 Hz, 2H, arom.CH), 7.60 (t, 1H, arom.CH), 7.93 (d, J = 8.5 Hz, 2H, arom.CH), 8.10 (d, 1H, arom.CH), 8.15 (s, 1H, H-4 of imidazole ring), 8.99 (s, 1H, CH=N) ppm. ^13^C-NMR (DMSO-d_6_) δ: 20.02 (CH_3_CO), 22.95 (CH_3_CO), 114.59 (C4 imidazole), 124.04 (C3 acetoxyphenyl), 126.18 (C1 acetoxyphenyl), 127.10 (C5 acetoxyphenyl), 127.38 (C5 imidazole), 127.85 (C2,6 chlorophenyl), 129.34 (C3,5 chlorophenyl), 132.84 (C6 acetoxyphenyl), 133.05 (C4 acetoxyphenyl), 148.60 (C1 chlorophenyl), 148.80 (C4 chlorophenyl), 150.36 (C=N), 156.28 (C2 acetoxyphenyl), 169.50 (C=O), 172.13 (C=O), 172.53 (C=S) ppm. Anal. Calcd for C_20_H_16_ClN_3_O_3_S (413.88). Calculated: C, 58.04; H, 3.90; N, 10.15. Found: C, 58.22; H, 4.07; N, 9.93.

### 3.5. Biological Studies

#### 3.5.1. Antitumor Activity against Three Cancer Cell Lines

Antitumor activity of the newly prepared imidazole compounds **4a**–**e** and **5** were carried out against MCF-7, HepG2, HCT-116, and MCF-10A using the MTT assay method. Cells at a density of 1 × 10^4^ were seeded in 96-well plates at 37 °C for 48 h under 5% CO_2_. After incubation, the cells were treated with different concentrations of the prepared molecules and incubated for 24 h. MTT dye was added at the end of 24 h of drug treatment and incubated for 4 h at 37 °C. Next, 100 µL of dimethyl sulfoxide was added to each well to dissolve the purple formazan formed. The color intensity of the formazan product, which represents the growth condition of the cells, is quantified using an ELISA plate reader at 570 nm. The experimental conditions were carried out with at least three replicates and the experiments were repeated at least three times.

#### 3.5.2. Cell Cycle Analysis of Compounds **4d** and **5**

MCF-7 cells (2 × 10^5^/well) were harvested and washed twice in PBS. After that, cells were incubated at 37 °C and 5% CO_2_. The medium was replaced with DMSO (1% *v/v*) containing the tested compounds, then incubated for 48 h, washed twice in PBS, fixed with 70% ethanol, rinsed again with PBS, and then stained with DNA fluorochrome PI for 15 min at 37 °C. DNA content was analyzed by flow cytometry on a FACS Calibur flow cytometer (Becton and Dickinson, Heidelberg, Germany).

#### 3.5.3. Annexin V FITC/PI Apoptosis Detection Staining Assay

Apoptosis in MCF-7 cells was investigated using fluorescent Annexin V-FITC/PI detection kit by flow cytometry assay. Briefly, MCF-7 cells (2 × 10^5^) after incubation for 12 h. Cells were treated with compounds **4d** and **5** at their IC_50_ concentration for 48 h, then the cells were harvested and stained with Annexin V-FITC/PI dye for 15 min in the dark at 37 °C. Flow cytometry analyses were performed using a FACS Calibur flow cytometer (Becton and Dickinson, Heidelberg, Germany).

#### 3.5.4. In Vitro VEGFR-2 Kinase Assay

VEGFR-2 inhibitory activity for compounds **4d**, **5,** and erlotinib was evaluated using human VEGFR-2 ELISA kits according to the manufacturer’s instructions. A total of 100 µL of standard solution and tested molecules were pipetted into the 96-wells and VEGFR-2 was bound to the wells by the immobilized antibody. The wells were washed and biotinylated antihuman VEGFR-2 antibody was added. After washing away unbound biotinylated antibody, 100 µL of conjugated streptavidin solution was pipetted into the wells. After washing, 100 µL of TMB substrate solution was added to the wells for 30 min at 37 °C. Stop solution (50 µL) was added, and the intensity of the color was measured immediately at 450 nm.

#### 3.5.5. In Vitro B-Raf Kinase Assay

RAF inhibitory activity for compounds **4d, 5,** and erlotinib was evaluated using human B-RAF ELISA kits according to the manufacturer’s instructions. Next, 100 µL of standard solution and tested molecules were pipetted into the 96-wells and B-RAF was bound to the wells by the immobilized antibody. The wells were washed and biotinylated antihuman B-Raf antibody was added. After washing away unbound biotinylated antibody, 100 µL of conjugated streptavidin solution was pipetted into the wells. After washing, 100 µL of TMB substrate solution was added to the wells for 30 min at 37 °C. Stop solution (50 µL) was added, and the intensity of the color is measured immediately at 450 nm.

### 3.6. Molecular Docking Study

A molecular docking study was performed using the MOE software program (MOE 2009.10). The VEGFR-2 crystal structure (PDB code: 3U6J) was obtained from a protein data bank (Appendix A).

## 4. Conclusions

In conclusion, a novel series of imidazoles bearing arylidene amino substituents at the *N*-1 position were synthesized in order to investigate their anticancer activity. The structure of the prepared imidazole derivatives was confirmed utilizing spectral data such as IR, ^1^H-NMR, and ^13^C-NMR along with elemental analyses. The prepared imidazoles molecules were screened for their cytotoxic activity against three cancer cell lines namely, MCF-7 breast cancer, HepG2 liver cancer, and HCT-116 colon cancer cell lines. Compound **5** was the most potent of all against the three cell lines. DNA flow cytometric analysis showed that, compounds **4d** and **5** exhibit pre-G1 apoptosis and cell cycle arrest at the G2/M phase. The results of the VEGFR-2 and B-Raf kinase inhibition assays revealed that compounds **4d** and **5** displayed good inhibitory activity compared with the reference drug erlotinib. Finally, imidazole compounds **4d** and **5** could serve as lead chemical entities for further structural optimization as anticancer agents.

## Data Availability

MDPI Research Data Policies” at https://www.mdpi.com/ethics.

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
