# Peer review of "5-Aryl-1-Arylideneamino-1H-Imidazole-2(3H)-Thiones: Synthesis and In Vitro Anticancer Evaluation"

_molecules, 2021, doi:10.3390/molecules26061706_

Round 1
Reviewer 1 Report
The Authors describe a synthesis of a set of compounds, which they most likely mistake to be hydrazones derived from 5-aryl-imidazolethione. The products were found to have antiproliferative and apoptosis – inducing effects, as well as displayed inhibition of VEGFR-2 and to some extent B-Raf enzymatic activities. For compound 5, the IC50 values for tumor cell lines approached 1 µM while the cytotoxic effect on normal cells was seen at >20 µM. Some idea on the mode of action was gathered from a series of FACS experiments. Interesting properties were found, particularly for compound 5, rather than the compounds of the series 4a-e. It is not clear why no series of compounds similar to 5 was neither made nor investigated. Inhibition of B-Raf was observed at a concentration of around 1 µM. This seems moderate.
The main issue the Reviewer has with this paper is that the structure of none of the products were adequately confirmed. The Reviewer believes that the unprecedented cyclization described in this paper allegedly leading to imidazole-2-thione is actually a well-known Hantzsch thiazoline synthesis. The products would be derivatives of 4-aryl-1,3-thiazol-2-ylhydrazine. The reviewer supports this statement by identical spectral data reported for isomeric products possessing thiazole ring (e.g., compare data for compound 4d with data of compound 1Bb described in Eur. J. Org. Chem. 2014, 3387–3394 doi: 10.1002/ejoc.201402129 (Supporting Info)). Some products do not match the literature data. The authors need to convince the Reviewer that their products are actually imidazoles before publication of this article could be allowed.
Additional minor comments that may help the Authors publish their data after reevaluation of their structures:
Citation 18 does not contain the promised description of the synthesis of precursor compounds, instead references it further to a journal that the Reviewer cannot access (Journal of the Institution of Chemists (India). 1993;65:133–135).
The authors make no statement on the water solubility of the obtained compounds. No mention of any solvent (DMSO ?) can be found in the description of the biological experiments. Are the compounds well soluble in water (>1mg/ml)?
The reviewer finds that most Figures need some rework. Figure and Table captions need to be self-explanatory. The control experiments need to be defined each time: a negative control should be named. Abbreviations used in the Figures should be explained, for example, in the caption. Data of some bar graphs are entirely given in the main text. Can the authors justify the need for such Figures?
Specific comments:
Figure 2: Part A – the way the percentages are normalized prevents the sum to deviate from 100%, why does the scale reach 120%. Define CTRL.
Part B – the vertical axis seems to show a percentage of all cells, while the bars indicate the Pre-G1 cycle?
Part C - Deconvolution fits without experimental data have little value. Please overlay them with the original experimental data or replace them with a table. The technical quality of Figure 2 is insufficient for publication.
Figure 3: Part A: Vertical axis seems to show a percentage of all cells, otherwise the three apoptosis modes would add up to 100%. The same colored bars are likely meant to show negative control, 4d and 5. Provide adequate labeling.
Figure 4: What is the meaning of the vertical lines and asterisks above the bars in this graph?
Figure 5: What is the meaning of the vertical lines and asterisks above the bars in this graph? What is the physical meaning of the negative IC50 value? By measuring the length of SD bar on this graph for the positive control (erlotinib), I arrived at 0.02 ± 0.91 µg/ml. Relative SD value of nearly 5000% does not indicate a valid measurement.
The authors might consider using only bold Arabic numerals for compounds in the Figures rather than jargon such as “Comp 5” as well as using Greek letter µ (micro) instead of u in the Figures.
The results of docking scores are given with 0.0001 kcal/mol precision. Such precision has no physical sense. Reduce to 0.01 or 0.1 kcal/mol (k should not be capitalized). The Authors state that from the docking energies “it is obvious that compound 5 was more potent than compound 4d”. In the opinion of the Reviewer the energy difference of merely 0.9 kcal/mol together with the imperfect nature of docking computation are insufficient to draw any obvious conclusion.
Author Response
Response to Comments from Reviewer #1:
Comments and Suggestions for Authors
The Authors describe a synthesis of a set of compounds, which they most likely mistake to be hydrazones derived from 5-aryl-imidazolethione. The products were found to have antiproliferative and apoptosis – inducing effects, as well as displayed inhibition of VEGFR-2 and to some extent B-Raf enzymatic activities. For compound 5, the IC50 values for tumor cell lines approached 1 µM while the cytotoxic effect on normal cells was seen at >20 µM. Some idea on the mode of action was gathered from a series of FACS experiments. Interesting properties were found, particularly for compound 5, rather than the compounds of the series 4a-e. It is not clear why no series of compounds similar to 5 was neither made nor investigated. Inhibition of B-Raf was observed at a concentration of around 1 µM. This seems moderate.
We really appreciate very much the reviewer for the constructive comments. We appreciate the suggestions made to improve the manuscript. The comments are valuable for us and for the paper. We have addressed all the comments as below.
The main issue the Reviewer has with this paper is that the structure of none of the products was adequately confirmed. The Reviewer believes that the unprecedented cyclization described in this paper allegedly leading to imidazole-2-thione is actually a well-known Hantzsch thiazoline synthesis. The products would be derivatives of 4-aryl-1,3-thiazol-2-ylhydrazine. The reviewer supports this statement by identical spectral data reported for isomeric products possessing thiazole ring (e.g., compare data for compound 4d with data of compound 1Bb described in Eur. J. Org. Chem. 2014, 3387–3394 doi: 10.1002/ejoc.201402129 (Supporting Info)). Some products do not match the literature data. The authors need to convince the Reviewer that their products are actually imidazoles before publication of this article could be allowed.
You have raised an important point here. However, the reaction is performed in basic media at reflux temperature which catalyze the reaction through NH2 rather than S. Also, some of this work is similar to 2020 paper in the Journal of Archiv der pharmazie, http: 10.1002/ardp.202000121. Additionally, the product was confirmed via spectroscopic techniques like 13C-NMR which showed the peak of thiocarbonyl group (C=S) at 178.06-178.28 ppm, which should disappear if the thiazole analogue is formed. (Supporting information).
Additional minor comments that may help the Authors publish their data after reevaluation of their structures:
Citation 18 does not contain the promised description of the synthesis of precursor compounds, instead references it further to a journal that the Reviewer cannot access (Journal of the Institution of Chemists (India). 1993; 65:133–135).
Another reference is added and it is highlighted, and the order of the references in the manuscript is corrected.
The authors make no statement on the water solubility of the obtained compounds. No mention of any solvent (DMSO?) can be found in the description of the biological experiments. Are the compounds well soluble in water (>1mg/ml)?
The compounds are soluble in DMSO and it is presented in the experimental part and highlighted. In addition, the compounds were tested for their water solubility and the results showed no water solubility at any concentration.
The reviewer finds that most Figures need some rework. Figure and Table captions need to be self-explanatory. The control experiments need to be defined each time: a negative control should be name. Abbreviations used in the Figures should be explained, for example, in the caption. Data of some bar graphs are entirely given in the main text. Can the authors justify the need for such Figures?
All the Figures and Tables are revised and improved through the manuscript.
Control experiments are defined
The negative control is named
We use Figure as it gives more efficient visual presentations of the findings.
Specific comments:
Figure 2: Part A – the way the percentages are normalized prevents the sum to deviate from 100%, why does the scale reach 120%. Define CTRL.
The scale of 120% is from automatic mode of the program and we apologize for that mistake. The vertical axis scale is corrected to 100%, and CTRL is defined.
Part B – the vertical axis seems to show a percentage of all cells, while the bars indicate the Pre-G1 cycle?
The vertical axis showed the % of pre-G1 phase and it is defined in the Figure.
Part C - Deconvolution fits without experimental data have little value. Please overlay them with the original experimental data or replace them with a table. The technical quality of Figure 2 is insufficient for publication.
Thank you for this suggestion. However, the Deconvolution fits showed the different peaks of each phase of the cell cycle profile. In addition, it contains the quantitative percentage of each phase of the cell cycle (on the right side of the graph C).
Figure 3: Part A: Vertical axis seems to show a percentage of all cells, otherwise the three apoptosis modes would add up to 100%. The same colored bars are likely meant to show negative control, 4d and 5. Provide adequate labeling.
We thank the reviewer for the comment. The Figure is improved and adequate labeling of the compound is added.
Figure 4: What is the meaning of the vertical lines and asterisks above the bars in this graph?
We apologize for the forgetfulness; the vertical line and asterisk definition are added
Figure 5: What is the meaning of the vertical lines and asterisks above the bars in this graph? What is the physical meaning of the negative IC50 value? By measuring the length of SD bar on this graph for the positive control (erlotinib), I arrived at 0.02 ± 0.91 µg/ml. Relative SD value of nearly 5000% does not indicate a valid measurement.
We thank the reviewer for the comment. There is error occurs during drawing of the Figure (the value of SD is replaced with the value of IC50). The vertical line and asterisk definition are added. We apologize for the error.
The authors might consider using only bold Arabic numerals for compounds in the Figures rather than jargon such as “Comp 5” as well as using Greek letter µ (micro) instead of u in the Figures.
We thank the reviewer for the comment. All the the Figures are improved and µ (micro) instead of u in the Figures is added.
The results of docking scores are given with 0.0001 kcal/mol precision. Such precision has no physical sense. Reduce to 0.01 or 0.1 kcal/mol (k should not be capitalized). The Authors state that from the docking energies “it is obvious that compound 5 was more potent than compound 4d”. In the opinion of the Reviewer the energy difference of merely 0.9 kcal/mol together with the imperfect nature of docking computation are insufficient to draw any obvious conclusion.
Thank you for pointing this out. We agree with this comment, docking scores are reduced to 0.01 kcal/mol (with the correction of k) and it is highlighted and the term “obvious” is removed

Reviewer 2 Report
5-Aryl-1-arylideneamino-1H-imidazoles-2(3H)-thione: Synthesis and In Vitro Anticancer Evaluation
Referee’s Report
The manuscript under revision describes the synthesis of a number of previously unreported imidazole thiones and the relevantanticancer evaluation.
This referee will only provide reviewing of the synthetic section of the manuscript, having not enough competence to evaluate the pharmacology tests.
Some corrections are needed before publication, as indicated below.
1] TITLE The name of the final heterocycles is not correct! The name shohld be “5-Aryl-1-benzylideneamino-1H-imidazole-2(3H)-thiones”.
2] The name of 3a-e is cited several times in the text: it should be “2-Benzylidenehydrazine-1-carbothioamides" but for 3e, which does not share the same general name and should actually be named separatelyntly as 2-(3-phenylallylidene)hydrazine-1-carbothioamide; furthermore appropriate stereochemistry (E/Z) should be provided in the name!!!
All over in the text:
- a) it seems useless to repeat the full name and just “3a-e” could be cited!
- b) “arylidene” should be accordingly corrected.
3] Also structures 1a-e, 3a-e, and 4a-e of Scheme 1 do not seem appropriate (“n” = CH=CH ???). This referee would suggest distinct structures for a-d and for e.
4] In the introduction, “Indazole is an important class” to be corrected as “Indazoles are an important class”
5] In the introduction, the sentence “with substituted phenyl groups such as chlorine or methoxy groups” should be replaced with “phenyl groups carrying a chloro or a methoxy substituent”.
Experimental - Compound description and identification
Generally speaking, compounds are rather well described. Just a few corrections/comments:
- Bottom of page 8: “drops of” to be removed
- In the 1H NMR descriptions, for all signals defined “m”, please cite the lower and the higher value in such order;
- Page 9, 4.2.1 (3a): please describe separately the two signals 8.24 and 11.47;
- Page 9, 4.2.2 (3b): if a signal is described as “d” or “t”, the J must be provided; it seems unlikely that the signal at 7.91 could be a singlet; signals at 8.15 and 8.38 are possibly inverted? Please verify;
- Page 9, 4.2.4 (3d): in the 13C spectrum, one carbon is missing. Perhaps two carbons are accidentally isochronous? The AA should declare this;
- Page 10, 4.3.5 (4e): the attribution of some signals is rather unconvincing, mainly for signals at 6.94, 7.88, 7.93. Have the AA ascertained such an attribution with specific experiments or it is just a tentative attribution? The signal at 7.02, being a “m”, should be described as an interval.
Some editing is needed throughout also to eliminate some duplications of words.
Author Response
Response to Comments from Reviewer #2:
5-Aryl-1-arylideneamino-1H-imidazoles-2(3H)-thione: Synthesis and In Vitro Anticancer Evaluation
Referee’s Report
The manuscript under revision describes the synthesis of a number of previously unreported imidazole thiones and the relevant anticancer evaluation.
We sincerely thank the reviewer for constructive criticisms and valuable comments, which were of great help in revising the manuscript. We have addressed all the editor and reviewer comments as below.
Some corrections are needed before publication, as indicated below.
1] TITLE The name of the final heterocycles is not correct! The name shohld be “5-Aryl-1-benzylideneamino-1H-imidazole-2(3H)-thiones”.
Corrections were done and it is highlighted.
2] The name of 3a-e is cited several times in the text: it should be “2-Benzylidenehydrazine-1-carbothioamides" but for 3e, which does not share the same general name and should actually be named separatelyntly as 2-(3-phenylallylidene)hydrazine-1-carbothioamide; furthermore appropriate stereochemistry (E/Z) should be provided in the name!!!
Thank you for this suggestion, the benzylidene is given for benzene ring, but in the present paper we use derivatives of benzene so the term arylidene is more general.
Appropriate stereochemistry (E/Z) are provided in the IUPAC name and it is highlighted.
All over in the text:
- a) it seems useless to repeat the full name and just “3a-e” could be cited!
Done
- b) “arylidene” should be accordingly corrected.
Thank you for this suggestion, however arylidene is more general as we use substituted aldehydes
3] Also structures 1a-e, 3a-e, and 4a-e of Scheme 1 do not seem appropriate (“n” = CH=CH ???). This referee would suggest distinct structures for a-d and for e.
We put (“n”= CH=CH) in order to give more efficient presentation to differentiate between the different aldehydes used and to give only one Scheme.
4] In the introduction, “Indazole is an important class” to be corrected as “Indazoles are an important class”
As suggested by the reviewer, the sentence is corrected
5] In the introduction, the sentence “with substituted phenyl groups such as chlorine or methoxy groups” should be replaced with “phenyl groups carrying a chloro or a methoxy substituent”.
As suggested by the reviewer, the sentence is corrected
Experimental - Compound description and identification
Generally speaking, compounds are rather well described. Just a few corrections/comments:
- Bottom of page 8: “drops of” to be removed
We apologize for mistake, the mistake was corrected
- In the 1H NMR descriptions, for all signals defined “m”, please cite the lower and the higher value in such order;
- Page 9, 4.2.1 (3a): please describe separately the two signals 8.24 and 11.47;
Done
- Page 9, 4.2.2 (3b): if a signal is described as “d” or “t”, the J must be provided; it seems unlikely that the signal at 7.91 could be a singlet; signals at 8.15 and 8.38 are possibly inverted? Please verify;
We thank the reviewer for the comment, the J values are added.
After revising the data, the peak at 7.91 ppm appears as singlet and integrating two protons.
Signals at 8.15 and 8.38 ppm are corrected, we apologize for the mistake.
- Page 9, 4.2.4 (3d): in the 13C spectrum, one carbon is missing. Perhaps two carbons are accidentally isochronous? The AA should declare this;
We apologize for the mistake, the peak is added and it is highlighted
- Page 10, 4.3.5 (4e): the attribution of some signals is rather unconvincing, mainly for signals at 6.94, 7.88, 7.93. Have the AA ascertained such an attribution with specific experiments or it is just a tentative attribution? The signal at 7.02, being a “m”, should be described as an interval.
The interval of multiplet signal is added and it is highlighted.
The attribution of the signals are based on experiments presented on similar work in the Journal of Archiv der pharmazie with imidazole-2-thione molecules containing phenylallylidene moiety, http: 10.1002/ardp.202000121.
Some editing is needed throughout also to eliminate some duplication of words.
After revising our manuscript to address the reviewers’ comments, we have had it rechecked by a native speaker of English. As a consequence, some duplication of words is removed throughout the text. We hope that this revised manuscript meets your expectations.
Reviewer 3 Report
The work is interesting and well structured, with a synthetic part, biological activity and docking. However, it has some aspects that need to be improved:
a.- In the Abstract Section, the 4d and 5 tags do not indicate anything with respect to the structures and therefore must be replaced by some structural facts that define them.
b.- Sorafenib is an international nonpropietary name and accordingly, has to be written with the first lowercase letter (see Introduction Section).
c.- On page 2, in "Finally, Full details about the synthesis, evaluation of the antitumor activity in vitro are reported herein" "Full" has to be written as "full".
d.- "Propidium iodide" (page 5) must be written as "propidium iodide".
e.- In compounds 4a, 4b,...(Experimental Part) "H", included in their IUPAC names, must be written in italics.
f.- In "Cell cycle analysis compound 4d and 5" (page 12), 4d and 5 must be written in bold characters.
Author Response
Response to Comments from Reviewer #3:
The work is interesting and well structured, with a synthetic part, biological activity and docking. However, it has some aspects that need to be improved:
We sincerely thank the reviewer for constructive criticisms and valuable comments, which were of great help in revising the manuscript. We have addressed all the editor and reviewer comments as below.
a.- In the Abstract Section, the 4d and 5 tags do not indicate anything with respect to the structures and therefore must be replaced by some structural facts that define them.
Thank you for this suggestion. Some structural fact is added and it is highlighted.
b.- Sorafenib is an international nonpropietary name and accordingly, has to be written with the first lowercase letter (see Introduction Section).
Revised as requested
c.- On page 2, in "Finally, Full details about the synthesis, evaluation of the antitumor activity in vitro are reported herein" "Full" has to be written as "full".
We thank the reviewer for observation. The mistake was corrected and highlighted
d.- "Propidium iodide" (page 5) must be written as "propidium iodide".
Revised as requested and it is highlighted
e.- In compounds 4a, 4b,...(Experimental Part) "H", included in their IUPAC names, must be written in italics.
We thank the reviewers for comment, all the compounds names through the manuscript are revised and "H", is written in italics.
f.- In "Cell cycle analysis compound 4d and 5" (page 12), 4d and 5 must be written in bold characters.
Done
Round 2
Reviewer 1 Report
The authors have not convinced me of their structures. There must be either an independent literature precedent or a definitive resolution of at least a single structure (e.g. X-ray).
The reviewer is aware that there are significant differences in the data published for isomeric thiazoles in 13C and mp. The authors make a point with their signal at ca. 180 ppm possibly attributed to C=S carbon. On the other hand, for 4,5-dimethylimidazole-2-thiones, the observed value for C-2 is in the 155-160 ppm range. The question of 4 vs 5 substitution is also not clear.
As to the minor comments, the Authors in their hasty revision did not replace all "u" for micro. The asterisks denoting statistical significance most likely refer to the positive control, do they? If that is the case, why there is more statistical power in comparison with less differing values (Fig. 4). There is no mention in the manuscript that the samples were introduced in DMSO. Only dissolution of formazan is now described to require DMSO.
Author Response
The authors have not convinced me of their structures. There must be either an independent literature precedent or a definitive resolution of at least a single structure (e.g. X-ray). The reviewer is aware that there are significant differences in the data published for isomeric thiazoles in 13C and mp. The authors make a point with their signal at ca. 180 ppm possibly attributed to C=S carbon. On the other hand, for 4,5-dimethylimidazole-2-thiones, the observed value for C-2 is in the 155-160 ppm range. The question of 4 vs 5 substitution is also not clear.
We thank the reviewer for the comments. Your scientific talk is correct, but before writing the manuscript we have reviewed many paper about thiazole and we don’t found the signal around 178 ppm so the peak at 178 ppm ppm possibly attributed to C=S carbon. Sadly, single crystal technique consumes time and also, we don’t have this facility in our university.
As to the minor comments, the Authors in their hasty revision did not replace all "u" for micro. The asterisks denoting statistical significance most likely refer to the positive control, do they? If that is the case, why there is more statistical power in comparison with less differing values (Fig. 4).
All "u" are replaced with micro (m).
After revising the data in Figure 4, there is one asterisk is added by mistake and the Figure is improved, we apologize for the mistake
There is no mention in the manuscript that the samples were introduced in DMSO. Only dissolution of formazan is now described to require DMSO.
A sentence regarding the solubility is added in the experimental part and it is highlighted.